

# RFPDR: a random forest approach for plant disease resistance protein prediction

Diego Simón[1,2,3], Omar Borsani[4] and Carla Valeria Filippi[4]

[1] Laboratorio de Virología Molecular, Centro de Investigaciones Nucleares, Facultad de Ciencias, Universidad de la República, Montevideo, Uruguay
[2] Laboratorio de Evolución Experimental de Virus, Institut Pasteur de Montevideo, Montevideo, Uruguay
[3] Laboratorio de Genómica Evolutiva, Departamento de Biología Celular y Molecular, Facultad de Ciencias, Universidad de la República, Montevideo, Uruguay
[4] Departamento de Biología Vegetal, Facultad de Agronomía, Universidad de la República, Montevideo, Uruguay

Corresponding author
Carla Valeria Filippi,
cfilippi@fagro.edu.uy

## ABSTRACT

**Background**. Plant innate immunity relies on a broad repertoire of receptor proteins that can detect pathogens and trigger an effective defense response. Bioinformatic tools based on conserved domain and sequence similarity are within the most popular strategies for protein identification and characterization. However, the multi-domain nature, high sequence diversity and complex evolutionary history of disease resistance (DR) proteins make their prediction a real challenge. Here we present RFPDR, which pioneers the application of Random Forest (RF) for Plant DR protein prediction.
**Methods**. A recently published collection of experimentally validated DR proteins was used as a positive dataset, while 10x10 nested datasets, ranging from 400-4,000 non-DR proteins, were used as negative datasets. A total of 9,631 features were extracted from each protein sequence, and included in a full dimension (FD) RFPDR model. Sequence selection was performed, to generate a reduced-dimension (RD) RFPDR model. Model performances were evaluated using an 80/20 (training/testing) partition, with 10-cross fold validation, and compared to baseline, sequence-based and state-of-the-art strategies. To gain some insights into the underlying biology, the most discriminatory sequence-based features in the RF classifier were identified.
**Results and Discussion**. RD-RFPDR showed to be sensitive ($86.4 \pm 4.0\%$) and specific ($96.9 \pm 1.5\%$) for identifying DR proteins, while robust to data imbalance. Its high performance and robustness, added to the fact that RD-RFPDR provides valuable information related to DR proteins underlying properties, make RD-RFPDR an interesting approach for DR protein prediction, complementing the state-of-the-art strategies.

## INTRODUCTION

Plants account for a double-layered immune system for protecting themselves against a broad spectrum of pathogens. The first layer of defense is composed by transmembrane
pattern recognition receptors (PRRs) that can recognize pathogen associated molecular patterns (also known as PAMPs), leading to a pattern-triggered immunity response (PTI, *Hofberger et al., 2014*). The PTI usually involves defense gene induction (*Chinchilla et al., 2006*), while lead cell wall reinforcement thought callose deposition, and the secretion of antimicrobial compounds (*Zipfel & Robatzek, 2010*).

Successful pathogens overcoming PTI can face the second layer of defense. This second layer is composed mostly by intracellular nucleotide-binding, leucine-rich repeat (NLR) type proteins. NLRs can recognize specific pathogen effectors, leading to a gene-for-gene resistance, known as effector-triggered immunity (ETI). Sometimes, pathogens overcome ETI by gaining new effectors or losing the recognizable ones (*Asai et al., 2018*); but some plants can also evolve to recognize new effectors, re-establishing ETI and returning to a resistant state (*Lu & Tsuda, 2020*). This 'zigzag model' (*Jones & Dangl, 2006*) summarizes the evolutionary dynamics between host and pathogen, and allows us to understand why NLRs are one of the most variable protein families. NLRs diversity relies not only on their extraordinarily polymorphic nature, but also on their presence-absence and copy-number variation, observed even between closely related individuals (*Van de Weyer et al., 2019*).

PRRs, including receptor-like proteins (RLP) and receptor-like kinases (RLK) lack a unifying conserved architecture. On the other hand, NLRs are usually described as multi-domain proteins with a highly conserved architecture, including a variable N-terminal domain (with a coiled-coil domain or a Toll/interleukin-1 receptor), a central nucleotide-binding (NB) domain, and a C-terminal leucine-rich repeat (LRR) domain (*Cesari et al., 2014*). Nevertheless, recent studies suggest that NLRs are more diverse, not only in terms of sequence, but also in terms of structure and activity, than previously thought (*Cesari, 2018*; *Barragan & Weigel, 2021*).

Characterizing the complete repertoire of these plant disease resistance (DR) proteins has been one of the main objectives of the plant research and breeding community (*Hofberger et al., 2014*). Nevertheless, their multi-domain nature, high sequence diversity and structural complexity, make DR proteins prediction a real challenge (*Kourelis & Kamoun, 2020*). Some bioinformatic tools have been developed for DR protein annotation, mostly based on conserved domain or sequence similarity (*Meyer et al., 2003*; *Steuernagel et al., 2015*; *Li et al., 2016*; *Osuna-Cruz et al., 2018*; *Santana Silva & Micheli, 2020*; *Toda et al., 2020*).

However, the main drawback of those strategies is that when the relevant signals are too weak to be detected by general consensus, protein annotation may fail. By contrast, machine learning (ML) classifiers, as random forest, can learn specific 'DR protein rules' from positive and negative training datasets without doing previous assumptions. Moreover, they can extract relevant signatures that are hidden in the sequence data (*Sperschneider et al., 2016*; *Sun et al., 2020*).

The success of a ML strategy relies on the availability of reference datasets. But, curated collections of experimentally validated sequences are scarce, even for widely studied protein families (*Kourelis & Kamoun, 2020*). Indeed, until recently PRGdb (http://prgdb.org/prgdb/) was the most comprehensive DR database, with only 153 reference proteins (including both PPRs and NLRs, *Osuna-Cruz et al., 2018*). This fact definitely limited the development of ML strategies for DR protein prediction, evidenced by only two reports until now

(*Pal, Jaiswal & Chauhan, 2016*; *Kushwaha et al., 2016*). In mid-2020, RefPlantNLR, an experimentally validated reference dataset of 415 NLRs, was published (*Kourelis & Kamoun, 2020*), thus providing the necessary framework for the development of robust ML-based prediction strategies for DR proteins.

Here, we present RFPDR, the first random forest-based disease resistance protein predictor. RFPDR was trained using a range of feature categories, estimated on a broad set of validated DR proteins. Features included protein length, sequence-based composition estimates (amino acid, dipeptide and tripeptides), autocorrelation (normalized Moreau-Broto, Moran and Geary; *Kawashima et al., 2008*), composition/transition/distribution (CTD, *Dubchak et al., 1995*; *Dubchak et al., 1999*), and conjoint triad descriptors (*Shen et al., 2007*). Added to its predictive capacity, RFPDR provides valuable information related to DR protein subjacent properties, including their proper grammar. This way, RFPDR emerges as an attractive and powerful strategy for DR protein prediction, that complements the current state-of-the-art methods.

# MATERIALS & METHODS

## Positive and negative datasets construction

Reference DR proteins (i.e., cloned or experimentally validated) from 79 species (32 genera), were selected from bibliography: (a) *Osuna-Cruz et al. (2018)*, 153 DR proteins (including NLRs and PRRs); (b) *Tang, Wang & Zhou (2017)*, 58 PRRs; (c) *Kanyuka & Rudd (2019)*, 19 PRRs; (d) *Kourelis & Kamoun (2020)*, 415 NLR. In addition, a literature revision was performed, in order to broaden the number of validated DR which conform the positive dataset, with special emphasis on PRRs (Table S1). Clustering (0.95, CD-HIT, *Fu et al., 2012*) was used for redundancy removal, retaining a final number of 400 DR proteins for positive dataset construction (289 NLRs and 111 non-NLRs).

In parallel, the whole proteome FASTA sequence (*El-Gebali et al., 2019*) of six well annotated species (three monocots: *Hordeum vulgare*, *Oryza sativa japonica* and *Triticum aestivum*, and three dicots: *Arabidopsis thaliana*, *Glycine max* and *Solanum lycopersicum*) were retrieved from Ensembl Plants (https://plants.ensembl.org, release 47). Interproscan (*Jones et al., 2014*) was run for these proteomes, and sequences containing Pfam motifs associated with DR proteins (*Osuna-Cruz et al., 2018*) were removed, keeping an initial number of 529,203 potential non-DR proteins. As a second step, an annotation-based filter was applied, by removing all of the remaining proteins that were previously annotated as involved in the defense process. Clustering (0.50, word length 3, CD-HIT) was performed for redundancy removal. Finally, CD-HIT-2D (*Fu et al., 2012*) was used to compare the positive and negative datasets. Those sequences belonging to the negative dataset that showed more than 63% of similarity to one of the positive dataset were blasted, and eliminated if BLAST showed the best hit with proteins associated with defense processes. After removing proteins shorter than 113 amino acids (i.e., 113 was the minimum length observed in the positive dataset), and applying the 'sanity-check' routine implemented in the protr R package (*Xiao et al., 2015*), a final number of 64,024 non-DR proteins were kept for negative datasets construction. A custom R script was used for the generation

**Figure 1** Schematic representation of the steps followed for automatic disease resistance protein prediction.

of 10 nested random samplings (with ten replicates), ranging from 400 to 4,000 non-DR proteins (from ratio 1:1, completely balanced; to 1:10, totally unbalanced, positive:negative datasets, Fig. 1).

An Intel® Core™ i9-7960X CPU with 2.80 GHz and 128 GB (8x16GB with 2,666MHz), running Ubuntu 18.04.5 LTS (GNU/Linux 4.15.0-140-generic x86_64), was used in data modelling. The source code and datasets presented here are available at: https://github.com/cvfilippi/rfpdr.

## Feature extraction and selection

For each protein, the feature vectors were calculated using protr (*Xiao et al., 2015*): amino acid, dipeptide and tripeptide composition; autocorrelation (normalized Moreau-Broto, Moran and Geary; *Kawashima et al., 2008*); Composition, Transition, Distribution (CTD, *Dubchak et al., 1995*; *Dubchak et al., 1999*); and conjoint triad (*Shen et al., 2007*). In addition, sequence length was estimated using a custom script, summarizing a total of 9,631 protein features. All these features were used for full-dimension model estimation.

For dimension reduction, 100 random samplings of $n = 400$ sequences were performed on the negative dataset, and combined with the 400 positive sequences (generating 100 1:1 positive:negative datasets). In each dataset, the Mann–Whitney-Wilcoxon $U$-test (with Bonferroni correction, *Midway et al., 2020*) was carried out. All features that were consistently significant in all datasets were initially kept for reduced-dimension model estimation. A panel of highly correlated features, capturing the same information, were also removed from the final set of features used for reduced-dimension model estimation.

## Random Forest for Plant Disease Resistance (RFPDR) model estimation and performance computation

The randomForest R package (*Liaw & Wiener, 2002*) was used for RFPDR models estimation, based both on full-dimension (FD-RFPDR) and reduced-dimension (RD-RFPDR). Each replicate of the positive:negative datasets (ratio 1:1 to 1:10), for FD-RFPDR and RD-RFPDR, were 80/20 train/test partitioned (caTools R package, *Tuszynski, 2020*). For validation of training datasets, 10-fold cross-validation was performed (Fig. 1).

Performance metrics were defined from the confusion matrices, which summarize the results of the RF classification (*Silva et al., 2019*). Specificity, accuracy, precision, sensitivity (a.k.a. recall) and F1-score were estimated (Fig. 2). Moreover, the area under the Receiver Operating Curve (ROC) was also calculated. The R packages vioplot (*Adler & Kelly, 2020*)

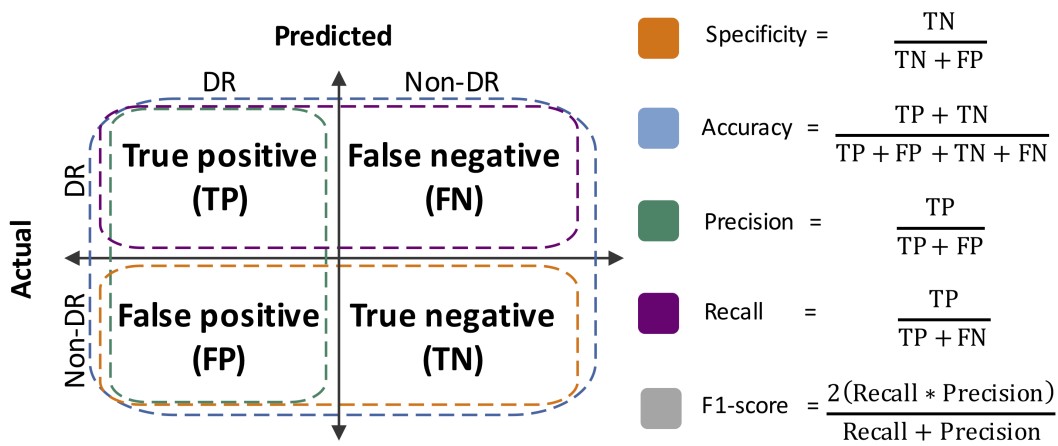

**Figure 2 Performance metrics estimated herein.** TN, true negative; FP, false positive; TP, true positive; FN: false negative.

and ROCR (*Sing et al., 2005*), and R-base functions (*R Core Team, 2020*), were used for statistical plots generation.

## Feature importance determination

The Gini-based importance (*Breiman et al., 1984*) was used for the determination of the most discriminant features for automatic disease resistance protein prediction. In this regard, the Gini-based importance score for the balanced (i.e., 1:1) FD-RFPDR and RD-RFPDR models were exploited. After ranking them, the 1% and 5%-top features, and the first features (which accounted for 50% of the importance), were further evaluated.

Moreover, and in order to get some insights into the DR proteins proper grammar, the 100 most discriminant amino acid, dipeptide and tripeptide composition features in the balanced (i.e., ratio 1:1) FD-RFPDR and RD-RFPDR models, were also individualized. Gini-based mean importance values were transformed into their proportional absolute frequencies, in order to generate word-cloud plots (wordcloud2, *Lang & Chien, 2018*).

## Comparison with sequence-similarity based, baseline and state-of-the-art strategies

The performance of one replicate 1:1 of our best RFPDR was compared against a sequence-similarity based strategy (PSI-BLAST, Position Specific Iterated BLAST, *Altschul, 1997*), a baseline method (Supported Vector Machine, SVM) and two state-of-the-art DR protein predictors, for the annotation of an independent test dataset accounting for 160 proteins (i.e., 20% test partitioned for the selected 1:1 RFPDR).

PSI-BLAST (5 iterations) was run locally, using for database construction all the positive and negative sequences, except those used as query. For protein prediction, the sequence accounting for the least e-value and highest score was selected for each query: if it was a DR sequence, the query was annotated as DR; otherwise, the query protein was annotated as non-DR.

**Table 1  List of 9,631 sequence-derived features.**

| Features (classes) | No. of extracted features FD-RFPDR | No. of extracted features RD-RFPDR |
|---|---|---|
| Sequence length | 1 | 1 |
| Amino acid composition | 20 | 4 |
| Dipeptide composition | 400 | 95 |
| Tripeptide composition | 8,000 | 646 |
| Normalized Moreau-Broto autocorrelation | 240 | 139 |
| Moran autocorrelation | 240 | 22 |
| Geary autocorrelation | 240 | 23 |
| Composition | 21 | 10 |
| Transition | 21 | 11 |
| Distribution | 105 | 18 |
| Conjoint triad | 343 | 164 |
| **Total** | **9,631** | **1,133** |

Regarding baseline methods, SVM was trained using the most significant features, on the same training sequences used for RFPDR model construction. For comparison purposes with previously reported ML strategies for DR protein prediction, the SVM was trained based on radial-basis function (RBF) kernel, using the R Packages caret (*Kuhn, 2008*) and e1071 (*Meyer et al., 2021*).

Finally, two state-of-the-art predictors, RRGPredictor (*Santana Silva & Micheli, 2020*) and DRAGO2 (*Osuna-Cruz et al., 2018*) were used for the annotation of the same independent test dataset. Both softwares were run locally, with *default* parameters.

Performance metrics (specificity, accuracy, precision, recall and F1-score) were estimated for all evaluated methods.

# RESULTS

## RFPDR model estimation and performance

A total of 9,631 sequence-derived features, belonging to 11 classes, were initially extracted, and used for FD-RFPDR. The Mann–Whitney-Wilcoxon $U$-test allowed the discrimination of the most significant features for DR protein identification (Table S2). After removing highly correlated features, a total of 1,133 out of 9,631 features were used for RD-RFPDR model estimation. The list and categories of features used for FD and RD-RFPDR is summarized in Table 1.

In order to test the effect of the unbalance of training datasets on the predictive performance, ten different FD-RFPDR and RD-RFPDR models were estimated (ratio 1:1 to 1:10, positive:negative datasets), yielding a total of 20 RFPDR (each one with 10 replicates). The performance metrics of each model are summarized in Fig. 3 and Table 2 (mean ± standard deviation). The corresponding ROC curves are presented in Fig. S1.

## Feature importance analysis

To analyze the importance of the features in the RFPDR classifier (balanced model, i.e., ratio 1:1 positive:negative), the Gini importance function was exploited (Table S3).

**Table 2  The performance metrics of each model (mean ± standard deviation).**

| Ratio positive: negative | Specificity | | Accuracy | | Precision | | Recall | | F1-Score | | AUC | | Time (m) | |
|---|---|---|---|---|---|---|---|---|---|---|---|---|---|---|---|
| | FD | RD | FD | RD | FD | RD | FD | RD | FD | RD | FD | RD | FD | RD |
| **RFPDR 1:1** | 96.4 ± 2.2 | 96.9 ± 1.5 | 92.0 ± 2.1 | 91.5 ± 2.3 | 96.2 ± 2.2 | 96.5 ± 1.9 | **87.7 ± 2.9** | 86.4 ± 4.0 | **91.8 ± 2.2** | 91.1 ± 2.5 | 96.7 ± 0.3 | 96.9 ± 0.3 | 3.39 ± 0.05 | **0.20 ± 0.01** |
| **RFPDR 1:2** | 99.0 ± 1.0 | 99.2 ± 0.6 | 92.0 ± 0.9 | 93.1 ± 1.6 | 97.8 ± 2.2 | 98.2 ± 1.6 | 79.2 ± 3.6 | 81.2 ± 4.0 | 87.4 ± 1.8 | 88.9 ± 2.7 | 96.8 ± 0.2 | 97.2 ± 0.4 | 5.34 ± 0.08 | 0.35 ± 0.02 |
| **RFPDR 1:3** | 99.5 ± 0.4 | 99.6 ± 0.4 | 95.0 ± 1.3 | 94.7 ± 0.7 | 98.1 ± 1.4 | 98.2 ± 1.8 | 81.3 ± 4.6 | 79.4 ± 1.7 | 88.8 ± 2.8 | 87.8 ± 1.4 | 96.7 ± 0.3 | 97.3 ± 0.3 | 7.54 ± 0.17 | 0.55 ± 0.03 |
| **RFPDR 1:4** | 99.8 ± 0.3 | 99.8 ± 0.2 | 95.2 ± 0.5 | 95.2 ± 1.2 | 99.0 ± 1.2 | 99.1 ± 1.1 | 76.6 ± 2.4 | 77.2 ± 4.4 | 86.3 ± 1.3 | 86.7 ± 2.9 | 97.1 ± 0.3 | 97.3 ± 0.4 | 9.64 ± 0.21 | 0.82 ± 0.01 |
| **RFPDR 1:5** | 99.8 ± 0.2 | 99.9 ± 0.2 | 95.8 ± 1.2 | 96.0 ± 1.1 | 98.8 ± 1.2 | 99.5 ± 1.1 | 76.7 ± 5.0 | 76.8 ± 5.5 | 86.3 ± 3.2 | 86.6 ± 3.4 | 96.8 ± 0.3 | 97.2 ± 0.2 | 12.40 ± 0.22 | 1.16 ± 0.02 |
| **RFPDR 1:6** | 99.9 ± 0.1 | 99.9 ± 0.2 | 96.8 ± 0.6 | 96.5 ± 0.8 | 99.5 ± 1.1 | 99.3 ± 1.4 | 77.3 ± 3.2 | 76.1 ± 4.2 | 87.0 ± 2.1 | 86.1 ± 2.7 | 96.8 ± 0.4 | 97.4 ± 0.2 | 15.06 ± 0.39 | 1.52 ± 0.03 |
| **RFPDR 1:7** | 99.9 ± 0.2 | 99.9 ± 0.1 | 96.6 ± 0.8 | 96.9 ± 0.8 | 99.0 ± 1.5 | 99.2 ± 1.3 | 74.8 ± 6.1 | 76.1 ± 6.2 | 85.1 ± 4.1 | 86.0 ± 4.2 | 96.9 ± 0.2 | 97.4 ± 0.4 | 18.16 ± 0.35 | 1.95 ± 0.05 |
| **RFPDR 1:8** | **100.0 ± 0.0** | 99.9 ± 0.1 | 97.2 ± 0.3 | 97.2 ± 0.4 | **100.0 ± 0.0** | 99.4 ± 0.8 | 75.2 ± 2.4 | 74.1 ± 3.9 | 85.8 ± 1.6 | 84.8 ± 2.3 | 96.8 ± 0.4 | **97.6 ± 0.4** | 21.67 ± 0.69 | 2.38 ± 0.05 |
| **RFPDR 1:9** | 100.0 ± 0.1 | **100.0 ± 0.0** | 97.6 ± 0.7 | 97.1 ± 0.6 | 99.7 ± 1.1 | 99.8 ± 0.6 | 75.6 ± 5.1 | 72.4 ± 4.0 | 85.9 ± 3.3 | 83.9 ± 2.6 | 96.9 ± 0.3 | 97.4 ± 0.3 | 25.87 ± 0.35 | 2.89 ± 0.09 |
| **RFPDR 1:10** | 100.0 ± 0.1 | 100.0 ± 0.1 | 97.5 ± 0.6 | **97.6 ± 0.5** | 99.6 ± 0.8 | 99.5 ± 0.8 | 73.2 ± 5.6 | 74.1 ± 4.6 | 84.3 ± 3.6 | 84.8 ± 3.0 | 96.9 ± 0.5 | 97.3 ± 0.3 | 30.72 ± 0.96 | 3.46 ± 0.13 |

**Notes.**

The best classification results on each model are underlined and in bold.

RFPDR, Random-Forest plant disease resistance; FD, full dimension; RD, reduced dimension; AUC, Area under the receiver operating curve; Time (m), computation time (minutes).
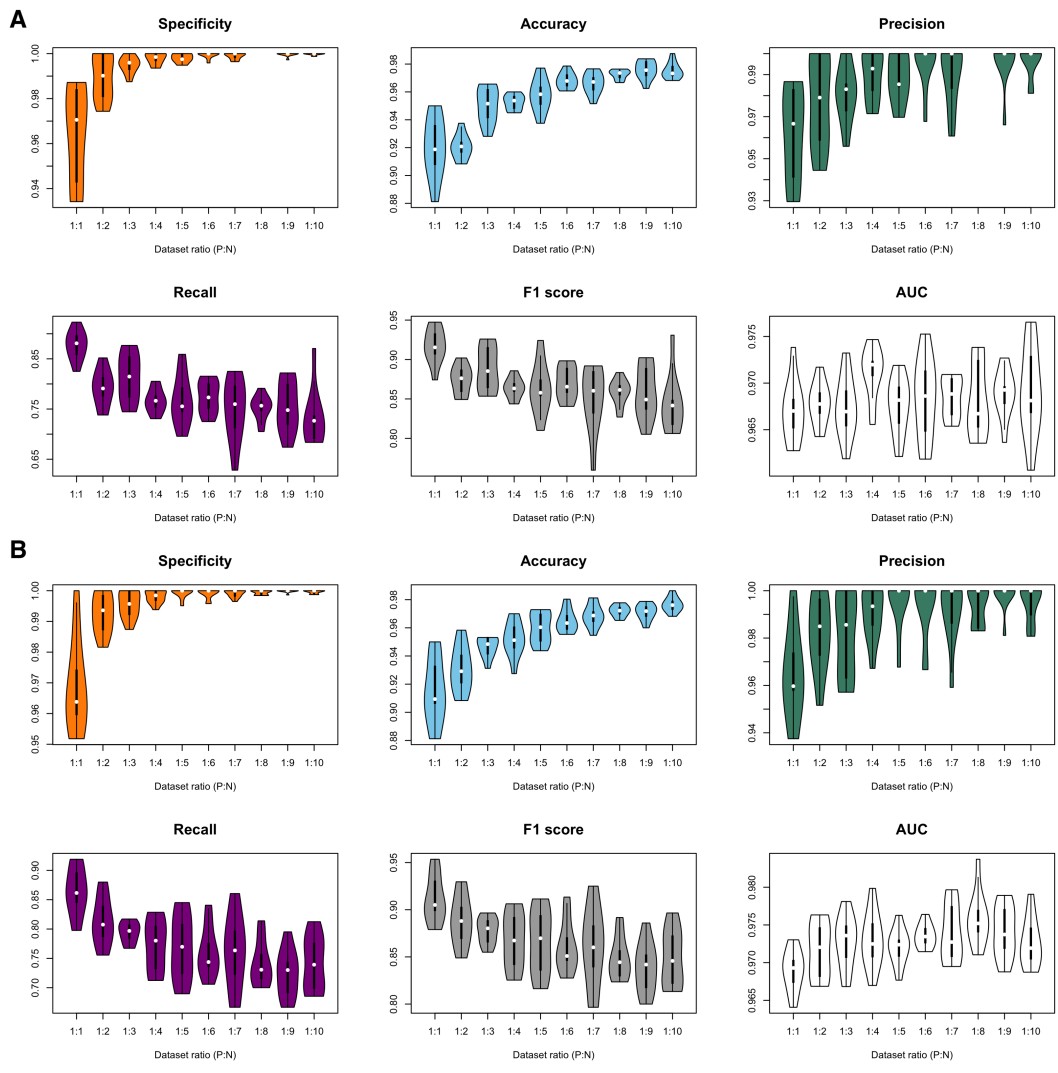

**Figure 3** **Violin plots of model performance metrics.** (A) Full-dimension random-forest plant disease resistance model (FD-RFPDR); (B) Reduced-dimension RFPDR (RD-RFPDR). AUC, Area under the receiver operating curve. P:N, positive:negative ratio.

Figure 4 (left) shows the ranking of the 5% top features by mean Gini importance value, and the cumulative importance score, in the FD-RFPDR (482 features) and the RD-RFPDR (57 features). Among models, the single feature importance was diverse and ranged from 1.31 to 0% in FD-RFPDR and from 3.44 to 0.005% in RD-RFPDR. Figure 4 (right) shows the importance order of the first features, which accounted for 50% of the importance in FD-RFPDR (213 features) and RD-RFPDR (90 features).

The categories of the most relevant features for each model were further inspected, and presented in Table 3. A further description of each feature can be accessed at Table S3. In addition, the top-1% features were used for a simplified model construction, in order to determine the pertinence of those features as specific for DR protein discrimination. The
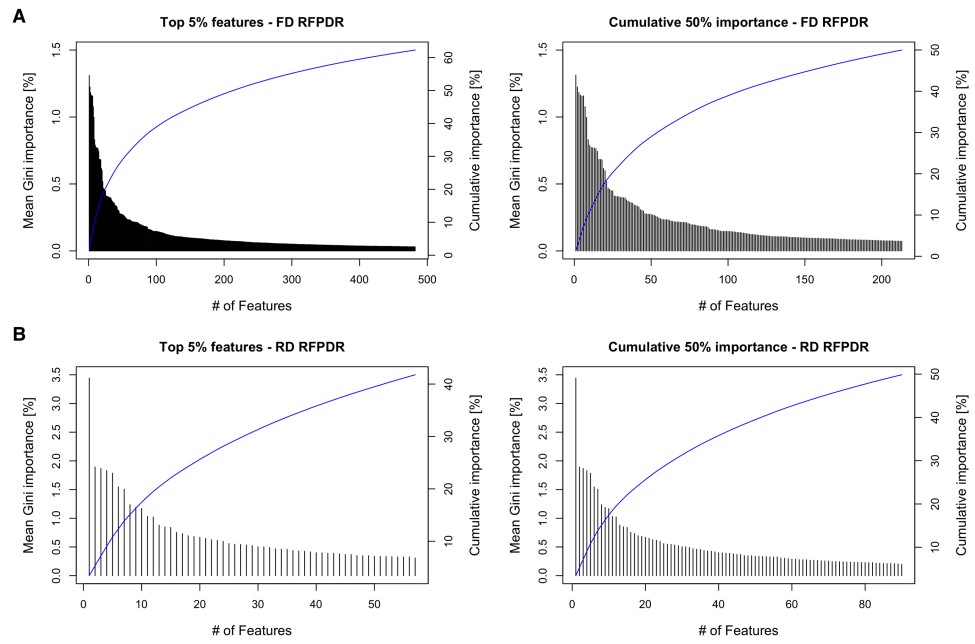

**Figure 4** **Gini-based importance feature-analysis.** (A) Full-dimension random-forest plant disease resistance model (FD-RFPDR); (B) Reduced-dimension RFPDR (RD-RFPDR. Left: Ranking orders of the 5% top features by importance value and their cumulative importance (blue line). Right: The importance of the top features, which accounted for 50% of the cumulative importance, and their cumulative importance (blue line).

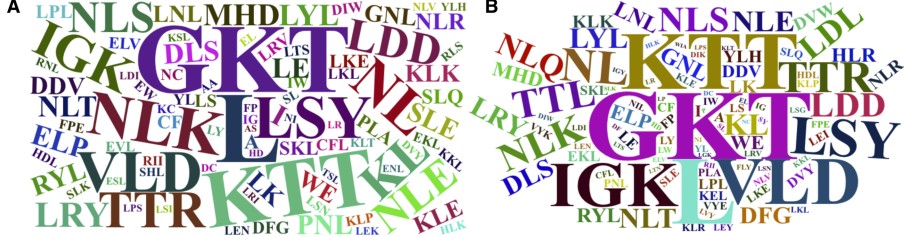

**Figure 5** **Word cloud plots with the 100-top most relevant amino acid, dipeptide and tripeptide composition features.** (A) Full-dimension random forest plant disease resistance model (FD-RFPDR); (B) reduced-dimension RFPDR (RD-RFPDR).

obtained metrics were specificity of 88.5 ± 3.3, accuracy 87.2 ± 3.1, precision 89.85 ± 3.6, recall 77.6 ± 3.6 and F1-score 83.3 ± 3.4.

The word cloud plots, presenting the 100-top most relevant amino acid, dipeptide and tripeptide composition features for FD-RFPDR and RD-RFPDR are presented in Fig. 5.

## Comparison with alternative strategies

Taking into account the results presented in Table 2, the RD-RFPDR models performed almost equal to the FD-RFPDR, but using a fraction of the features, thus avoiding potential overfitting. In addition, the RD model showed a significant reduction in processing times.

**Table 3 Categories of the most relevant features for each model.**

| Features (classes) | No. of most relevant features (FD-RFPDR) | | | No. of most relevant features (RD-RFPDR) | | |
|---|---|---|---|---|---|---|
| | Top 1% | Top 5% | Cumulative 50% | Top 1% | Top 5% | Cumulative 50% |
| Sequence length | 1 | 1 | 1 | 1 | 1 | 1 |
| Amino acid composition | 3 | 6 | 3 | 1 | 1 | 3 |
| Dipeptide composition | 5 | 68 | 24 | 0 | 4 | 8 |
| Tripeptide composition | 33 | 150 | 78 | 5 | 29 | 42 |
| Normalized Moreau-Broto autocorrelation | 10 | 106 | 30 | 2 | 6 | 12 |
| Moran autocorrelation | 3 | 18 | 7 | 0 | 2 | 3 |
| Geary autocorrelation | 4 | 19 | 8 | 0 | 3 | 4 |
| Composition | 8 | 15 | 9 | 1 | 5 | 8 |
| Transition | 5 | 12 | 8 | 2 | 4 | 5 |
| Distribution | 20 | 30 | 21 | 0 | 0 | 0 |
| Conjoint triad | 4 | 57 | 24 | 0 | 2 | 4 |
| **Total** | **96** | **482** | **213** | **12** | **57** | **90** |

**Notes.**

RFPDR, Random-Forest plant disease resistance; FD, full dimension; RD, reduced dimension.

**Table 4 Performance metrics –Comparison with other methods.**

| | RD-RFPDR | PSI-BLAST | SVM | RRGpredictor | DRAGO2 |
|---|---|---|---|---|---|
| Specificity | **1.000** | **1.000** | 0.986 | 0.986 | **1.000** |
| Accuracy | 0.950 | 0.799 | 0.906 | 0.963 | 0.969 |
| Precision | **1.000** | **1.000** | 0.987 | 0.988 | **1.000** |
| Recall | 0.911 | 0.640 | 0.844 | **0.944** | **0.944** |
| F1-Score | 0.953 | 0.781 | 0.910 | 0.966 | **0.971** |

**Notes.**

The best classification results on each model are underlined and in bold.

RD-RFPDR, reduced dimension - Random forest plant disease resistance predictor; PSI-BLAST, Position Specific Iterated BLAST (*Altschul, 1997*); SVM, Support Vector Machine; RRGPredictor, (*Santana Silva & Micheli, 2020*); DRAGO2, (*Osuna-Cruz et al., 2018*).

Here, the performance of one replicate of our RD-RFPDR 1:1 model, was compared to a sequence-similarity based method (PSI-BLAST), a baseline strategy (radial SVM, following the proposal of *Pal, Jaiswal & Chauhan (2016)* and two state-of-the-art predictors (DRAGO2, RRGPredictor, Table S4). The performance metrics for the classification of independent 160 sequences test dataset (i.e., 20% test partition of the data, for RD-RFPDR 1:1) reached by each strategy are summarized in Table 4.

# DISCUSSION

The potential and actual applications of machine-learning (ML) in plant biology have been exhaustively reviewed (*Silva et al., 2019*; *Mahood, Kruse & Moghe, 2020*). Independently of the field of application, or even the ML strategy, ML robustness strongly depends on the availability of confident data for model training. The increasing availability of plant sequencing data, including whole genomes, is opposed to the limited number of
known and well characterized DR proteins (i.e., cloned or experimentally validated). This important gap between sequence information and experimental validation has precluded the development of robust ML classifiers for DR protein prediction, with only two reports until now. One of them (DRPPP, *Pal, Jaiswal & Chauhan, 2016*) is a support vector machine (SVM) based tool, trained with only 112 validated DR proteins, including both NRLs and PRRs. The other one (NBSPred, *Kushwaha et al., 2016*), is also a SVM-based pipeline, that used predicted NLRs as a training dataset for NLR annotation. Unfortunately, none of them are available today, due to broken links.

During 2020, a comprehensive collection of experimentally validated NLRs with more than 400 sequences in 31 plant genera, was published (RefPlantNLR, *Kourelis & Kamoun, 2020*). This significant increase in validated DR proteins sequence availability, motivates the development of a new, and more robust, ML approach for DR protein prediction. In addition, here we complemented the available collection of known DR proteins (*Kanyuka & Rudd, 2019*; *Kourelis & Kamoun, 2020*; *Osuna-Cruz et al., 2018*; *Tang, Wang & Zhou, 2017*), which is remarkably enriched in NLRs, with a panel of carefully reviewed PRRs from bibliography (Table S1).

Beside ML approaches, random forest (RF) emerges as an attractive option for the current task, given that it is effective in binary classification problems (e.g., is a DR protein or not), is less computationally expensive, and also allows to understand how each variable (feature) is contributing to the prediction model.

Even though RF is more robust to overfitting than other ML strategies (*Breiman, 2001*; *Deneke, Rentzsch & Renard, 2017*), feature selection is still recommended (*Lv et al., 2019*). Here, feature selection was performed by using a non-parametric test (Bonferroni-corrected Mann–Whitney-Wilcoxon $U$-test), followed by elimination of some highly correlated features. As a result, feature counts were reduced from 9,631 to 1,133, significantly different between DR and non-DR proteins. This way, a full dimension RF model (named FD-RFPDR), based on 9,631 features and a reduced dimension RF model (named RD-RFPDR), based on 1,133 features, were developed and used in subsequent analyses.

RF are built on decision trees, which are sensitive to class imbalance. To test this issue, 10 FD- and 10 RD-RFPDR were built based on increasing unbalanced ratios, from 1:1 positive:negative sequences ratio (i.e., 400 positive and 400 negative sequences dataset, balanced), to 1:10 positive:negative (i.e., 400 positive and 4,000 negative sequences). Moreover, and given that RF if not deterministic, ten replicates of each model were done, in order to get a correct estimate of the performance metrics. Therefore, a total of 200 RF models were trained and tested here.

Independently of the imbalance ratio, the overall performance metrics were always high: specificity values ranged from $96.4 \pm 2.4$ to $100.0 \pm 0.1$; accuracy from $91.5 \pm 2.3$ to $97.6 \pm 0.5$; precision from $96.2 \pm 2.2$ to $100.0 \pm 0.0$; recall from $72.4 \pm 4.0$ to $87.7 \pm 2.9$; F1-score from $83.9 \pm 2.6$ to $91.8 \pm 2.2$ and AUC from $96.7 \pm 0.3$ to $97.6 \pm 0.4$ (Fig. 3, Table 2). When analyzing each model separately, the main impacts of the imbalance increase became apparent in specificity, accuracy, recall and F1-score metrics, while precision and AUC seemed unaffected. As expected, a higher proportion of the 'negative class' increases the specificity score, which is a measure of 'true negative' assignment.

Imbalance also has a positive impact on accuracy values, which take into accounts overall classification performance. This effect is also expected: the higher the imbalance, the higher the probability of assigning a correct value to the most abundant class, even by chance. On the other hand, the imbalance increase had a negative impact in recall and F1-score values. Recall measures the 'true positive' assignment (in this case, reference DR proteins that were correctly classified) while F1 is a more comprehensive score, as it combines precision and recall into one metric. In this regard, the balanced model (ratio 1:1) reached the higher performance for DR protein classification, in terms of recall and F1-score, and was selected for further analysis.

Something to highlight is that, when working with big data, computation time and complexity became key factors on determining the best bioinformatics approach and language (*Fourment & Gillings, 2008*). Beside models, RD-RFPDR outperformed FD-RFPDR, not only when considering performance metrics, but also by showing a significant processing time reduction. Moreover, dimension reduction also impacts on memory footprint requirements, pointing RD-RFPDR as our best model.

This fact was also apparent when analyzing the feature importance in the RFPDR classifier. The analysis of features by Gini importance value showed a significant number of features in FD-RFPDR with importance equal to zero, suggesting that they are not informative for DR protein prediction, while could increment noise. On the other hand, all the features used in RD-RFPDR model construction showed importance values different from zero. When analyzing the cumulative Gini importance score, more than twice features were needed in FD-RFPDR to reach the same 50% cumulative importance score. The model trained using only the top 1% features achieved a high performance, suggesting that those top 1% can be considered 'intrinsic features' for predicting DR from non-DR proteins.

To further analyze the feature biological meaning, the categories of the most important features were evaluated. From a numeric perspective, dipeptide and tripeptide composition, normalized Moreau-Broto autocorrelation, conjoint tried and composition (C) emerged as the most frequent categories (both in top 5% and cumulative 50% importance, for FD-RFPDR and RD-RFPDR) (Table 3). Conjoint triad and distribution (D) features were also abundant in FD-RFPDR, but not in RD-RFPDR. From the ranking perspective, sequence length (and features capturing length information, as D) emerge as the most important feature, followed by some transition (T) attributes related to polarity and hydrophobicity, normalized Moreau-Broto autocorrelation, selected tripeptides (GKT, KTT, VLD) and the leucine amino acid (Table S3). The fact that features capturing sequence length-related information emerge among the most important ones makes sense, given that DR proteins often contain tandem repeats (i.e., leucine-rich repeats, LRRs). Tandem repeats are present in the longer proteins, as has already been documented (*Delucchi et al., 2020*). From the three types of autocorrelation descriptors estimated herein (Moreau-Broto, Moran and Geary), normalized Moreau-Broto's were within the most discriminant features for DR protein classification. This could be a consequence of the definition of each descriptor: while Moreau-Broto uses the property values on measurement basis, Moran and Geary measure spatial-autocorrelation (*Ong et al., 2007*). From the different properties used for defining the autocorrelation descriptors, hydrophobicity emerged as the most discriminant

for DR protein classification. Hydrophobicity, added to polarity and normalized Van der Waals volume, were also among the most discriminant attributes, when considering C and T features (Table S3). The emergence of hydrophobicity as one of the most discriminant properties is in accordance with both PRRs and NLRs proteins structure and function, as described by *Morita et al. (2016)*, *Gómez-Gómez & Boller (2000)*; *El Kasmi & Nishimura (2016)*, beside others. Regarding their structure, N-terminus hydrophobic domains are consistent with signal peptides and transmembrane domains (*Gómez-Gómez & Boller, 2000*). Moreover, the LRR domains in LRR-RLKs were reported to be composed of parallel $\beta$ sheets, with their L residues facing towards the inner side of the molecule, generating a hydrophobic core (*Morita et al., 2016*). Hydrophobic residues were also reported to be required for NLR dimerization (*El Kasmi & Nishimura, 2016*). Moreover, the highly conserved Walker B motif, located in the nucleotide-binding site of NLRs, is characterized by the signature *hhhhDD/E* (where *h* represents a hydrophobic amino acid, *Proell et al., 2008*). When considering their function, DR receptors perceive and interact with signatures of pathogens and pests to initiate immune pathways. All these interactions are related to Van der Waals, electrostatic and hydrophobic interactions (*Bentham et al., 2017*; *Burdett et al., 2019*; *Wróblewski et al., 2018*). Regarding the last feature group evaluated herein, all the top important conjoint triads contained in their triad an amino acid belonging to the group {IFLP}, which includes leucine (Table S3).

Sequence-based composition importance was further investigated, in order to gain some insights into the underlying DR proteins proper grammar. Visual inspection of the word cloud plots generated using the 100-top sequence for both FD and RD-RFPDR, showed that the tripeptides GKT, followed by KTT, VLD, LSY, NLK, TTL, HRL, LDD, the single leucine amino acid (L) and other dipeptides containing L (KL and NL) emerge as the most important features. These results are in agreement with DR protein biological information. As mentioned before, NLRs, one of the most important DR protein classes (and the most abundant in our positive dataset), usually contain LRR motifs, with the conserved signature sequence *LxxLxLxxNxL*. In this regard, it is not surprising that L appeared as the main amino acid. GKT, the most discriminant tripeptide, and KTT, are conserved motif in NBS phosphate-binding loop (P-loop), the primary structure of which typically consists of a glycine-rich sequence followed by a conserved lysine and a threonine (*GxxGxGKTTx*). VLD and LDD are conserved motif in kinase-2 (*LxxLDDV* and *LvLDDvW*), present both in NLRs and receptor-like kinases (*Di Gaspero & Cipriani, 2003*; *Shimizu et al., 2015*). Remarkably, even though the aspartate-derived amino acids lysine (K) and a threonine (T) conform some of the most discriminant dipeptides and tripeptides, they were not significant as single amino acids.

In the comparison process with alternative and state-of-the art strategies, the least performance was achieved by the sequence-similarity based method, PSI-BLAST (Table 4, Table S4). Regarding SVM, even though the unavailability of the DRPPP model reported by *Pal, Jaiswal & Chauhan (2016)* precluded direct comparison, here we trained our SVM based on radial-basis kernel, as suggested by the authors. Our RD-RFPDR outperformed the SVM, reinforcing the notion that RF is a robust strategy for the current task. In addition, two state-of-the-art methods (DRAGO2, *Osuna-Cruz et al., 2018*, RRGPredictor, *Santana Silva*

*& Micheli, 2020*) were used for DR protein prediction in the same independent test-dataset. These two strategies, added to RGAugury (*Li et al., 2016*), are the unique reported methods for the identification of the full repertoire of DR proteins (i.e., NLRs and non-NLRs). Many other strategies exist, but focusing only on NLR prediction (e.g., *Steuernagel et al., 2015*; *Steuernagel et al., 2020*; *Toda et al., 2020*). DRAGO2 (*Osuna-Cruz et al., 2018*), based on sequence similarity and domain composition, consists of a Bash script; it requires a FASTA file as input, and is fast enough. RRGPredictor (*Santana Silva & Micheli, 2020*) is a Perl script that depends on Interproscan (*Jones et al., 2014*) output for DR protein prediction. RGAugury (not used here) presents a higher complexity, since it requires the installation of nine softwares, eight modules, four libraries, and three databases (*Li et al., 2016*). In contrast, RD-RFPDR can be used under all main operating systems, including Windows, as the source code is written in R. Regarding the comparative performance metrics obtained by our method with respect to the state-of-the-art strategies, our RD-RFPDR performed slightly better in some cases, and produced similar results in others. Even though we did not observe a significant performance increase, the fact that the proposed ML strategy does not rely on previous assumptions, as domain composition and/or sequence similarity, makes our RD-RFPDR highly flexible for DR protein prediction, even in rare or unknown species. The independence of domain composition in ML approaches can allow the identification of DR proteins that lack canonical or standard DR domains. Indeed, our RD-RFPDR was the unique strategy able to correctly classify Ty-1 (Tomato yellow leaf curl disease resistance gene Ty-1), a non-canonical DR (Table S4). In this regard, it is feasible that its prediction can be further enhanced in the future, when a larger number of experimentally validated DR proteins become available.

## CONCLUSIONS

In this work, we present the development of a plant disease resistance protein predictor, based on random forest (RFPDR). By using a comprehensive set of experimentally validated NLRs and PRRs, and an efficient feature selection process, RD-RFPDR emerged as our best model. RD-RFPDR showed to be sensitive and specific in identifying DR proteins, while robust to data imbalance. Moreover, the independence from sequence-similarity and/or domain composition gives RD-RFPDR flexibility, for the identification of DR proteins that lack the standard domains, while providing insight into the DR proteins underlying biology. These properties, added to the facts that our model is computationally efficient and can be used under different operating systems, make RD-RFPDR an attractive approach for DR protein prediction, complementing the current state-of-the-art strategies.

## ACKNOWLEDGEMENTS

This work used computational resources from Laboratorio de Genómica Evolutiva, Facultad de Ciencias, Universidad de la República, Uruguay. The authors are grateful to the two anonymous reviewers whose comments and suggestions have greatly improved the manuscript.

### Funding

Diego Simón and Carla Valeria Filippi are funded by Comisión Académica de Posgrado, Universidad de la República, Uruguay. The funders had no role in study design, data collection and analysis, decision to publish, or preparation of the manuscript.

### Grant Disclosures

The following grant information was disclosed by the authors:
Comisión Académica de Posgrado, Universidad de la República, Uruguay.

### Competing Interests

The authors declare there are no competing interests.

### Author Contributions

- Diego Simón conceived and designed the experiments, performed the experiments, analyzed the data, prepared figures and/or tables, authored or reviewed drafts of the paper, and approved the final draft.
- Omar Borsani conceived and designed the experiments, authored or reviewed drafts of the paper, and approved the final draft.
- Carla Valeria Filippi conceived and designed the experiments, performed the experiments, analyzed the data, prepared figures and/or tables, authored or reviewed drafts of the paper, and approved the final draft.

### Data Availability

The source code and datasets are available at GitHub: https://github.com/cvfilippi/rfpdr.

### Supplemental Information

Supplemental information for this article can be found online at http://dx.doi.org/10.7717/peerj.11683#supplemental-information.

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
