# Peer review of "RFPDR: a random forest approach for plant disease resistance protein prediction"

_PeerJ, doi:10.7717/peerj.11683_

## Round 0.1 · original submission · Major Revisions

Please carefully address all the critiques of the reviewers and revise your manuscript accordingly.

Reviewer 1 ·

Basic reporting

In general, the work is solid but routine judged from this type of work in the field of bioinformatics. The manuscript is fairly clear.

Experimental design

1) Some baseline methods (such as SVM, Logistic Regression) should be tested based on the dataset and feature vectors used in this work. By doing so, the readers will have global understanding of different ML algorithms in this topic.
2) It is also necessary to compare the proposed method against PSI-BLAST (i.e., Sequences in testing set should be PSI-BLASTed against training set. If the label of top hit is the same as the query sequence, a correct prediction is obtained; otherwise, it is a wrong prediction). By doing so, the readers will know the advantage of the proposed alignment free method in comparison to sequence alignment-based method.
3) The choice of P:N ratio seems to be an open issue in this type of bioinformatics task. In real application, which ratio should be used for training models?

Validity of the findings

No comment.

Additional comments

1) Line 98, the threshold used for sequence redundancy removal should be provided.
2) Line 153-158, the implementations of RRGPredictor and DRAGO2 should be provided. Were the benchmarking experiments done through the corresponding web servers or software for local use?
3) Table 4, what are the thresholds for these three methods? It seems that ROC curves or PRC curves are more informative to compare these three methods comprehensively.
4) Table 4, the computational time means the time for all predictions or one prediction? Moreover, the computer configuration should be mentioned when talking about the computational time.

Reviewer 2 ·

Basic reporting

More details in general comments section

Experimental design

More details in general comments section

Validity of the findings

More details in general comments section

Additional comments

In this manuscript, authors used a random forest approach to predict disease resistance (DR) genes in plants. This approach is novel, since previous studies were using SVM and using a much smaller training set. Overall, the analysis done is solid and the manuscript is well written. However, I am concerned whether ML is really needed, and whether existing approaches are good enough. There also needs to be a better discussion of the results on why some features are important. My detailed comments are below:

Major comments:
Introduction – authors should give more explanation for the diversity of DR genes. For an uninformed reader, this will not be clear. Also, is DR the same as Pathogen Recognition Genes (PRGs) used in Osuna-Cruz, 2018 and in PRGdb? Why not use the same abbreviation? If it is called DR, there are multiple mechanisms of disease resistance – are the authors trying to predict genes involved in all of these mechanisms? Clarify only.

Are all proteins in the non-DR set really non-DR? Authors only removed proteins with PFAM domains previously mentioned in Osuna-Cruz et al, 2017. Does that mean DR proteins can simply be found by searching for those PFAM domains? The problem of defining a true negative set for ML is a real serious problem in biological ML.

L164: Why was the Mann-Whitney-Wilcoxon test used only for tripeptides and not for other features? Appropriate feature selection must be performed even for other features, since there are too many features for the number of training examples, leading to overfitting. Authors should carefully consider if all these features are really needed for DR protein prediction.

Supplementary Table S1: Authors used an fdr p-value threshold of 0.05 to select tripeptide motifs. I think since the comparison is between 500 positive and 68,000 negative examples, having 5 instances in positive and 0 instances in negative is also going to show up as very significant. Authors should try to filter using more stringent threshold or some threshold based on frequency. Most features have minimal effect on model performance anyways.

Fig. 3/Table 2 – the high performance using all strategies indicates some existing trivial structure to the positive and negative training sets. I am not sure what it is, but my hunch is the filtering for PFAM domains that the authors did for the negative set. The positive set must then contain PFAM domains with unique structures not present in the negative set. Thus I feel the problem could be simply solved by knowing the domain architecture.

Fig. 4: To complement this analysis, authors should also look at the top 10-20 features and determine the performance metrics if only those features are used, and present it as Supplementary. As presented here, the analysis seems to be a black box, and why the top features are important is not clear. Authors provide some discussion on lines 280-288, but it is only for some tripeptide motifs. What about CTD, transition and composition, which are the most important feature sets?

On line 268, authors mention Moreau-Broto autocorrelation and conjoint triad as the most frequent categories. What are those? Why not some other autocorrelation? Looking at Table S2 further confused me on what the individual features really correspond to in terms of protein structure.

Table 4 – it is not clear if the performance of this method is really better than previous methods. Why not just use DRAGO2 or RRGPredictor? The authors should also compare their performance to that using simply PFAM/Interpro domain information. From the RRGPredictor paper, it appears that there are 10 standard domains in DR proteins. Would simply using Interproscan to find these domains be equally as good as this?

Are there new types of DR genes identified using this approach that RRGpredictor or PRGdb are not able to identify?

In the Discussion, authors mention that Interproscan takes time, but most genome databases already provide domain information. Softwares like hmmscan also run within a matter of minutes on entire plant proteomes using multi-processor architectures. Furthermore, I believe using RD-RFPDR would require calculating all features for all proteins for each genome using protr, which would also be a time-consuming step, comparable to hmmscan/IPRscan. So I am not sure if this is a valid argument.

Minor comments:
Abstract: While the rest of the manuscript is written properly, there are some grammatical mistakes in the abstract. E.g. “are among the”, “in order to generate also a”

Lines 27-224 where authors describe the importance of different features should be moved to the introduction. Why not also use domain-related features?

Fig. 3 is only briefly mentioned in the Results, and although these results are in Discussion, the figure is not cited. Please add the citation in Discussion.

---

## Round 0.2 · accepted · Accept

I am glad to let you know that both reviewers are satisfied with your revision and responses to their critiques.

Reviewer 1 ·

Basic reporting

No comment.

Experimental design

No comment.

Validity of the findings

No comment.

Additional comments

Thanks for the authors' efforts to carefully address my comments. I would like to suggest the acceptance.

Reviewer 2 ·

Basic reporting

The authors have addressed all my concerns, and the manuscript looks good. Despite Table S3, I would still add some information about the different features, especially the unusual autocorrelation features, in the Intro/Results/Discussion. I would also recommend splitting the Discussion into more paragraphs. These are optional recommendations.

Experimental design

No comment

Validity of the findings

No comment

Additional comments

No comments.